# *Faecalibacterium prausnitzii* Ameliorates Colorectal Tumorigenesis and Suppresses Proliferation of HCT116 Colorectal Cancer Cells

**DOI:** 10.3390/biomedicines10051128

**Published:** 2022-05-13

**Authors:** Ifeoma Julieth Dikeocha, Abdelkodose Mohammed Al-Kabsi, Hsien-Tai Chiu, Mohammed Abdullah Alshawsh

**Affiliations:** 1Faculty of Medicine, University of Cyberjaya, Persiaran Bestari, Cyberjaya 63000, Malaysia; ifeomadikeocha19@gmail.com (I.J.D.); abdelkodose@cyberjaya.edu.my (A.M.A.-K.); 2Department of Chemistry, National Cheng Kung University, Tainan City 70101, Taiwan; hchiu@mail.ncku.edu.tw; 3Department of Biochemistry and Molecular Biology, College of Medicine, National Cheng Kung University, Tainan City 70101, Taiwan; 4Department of Pharmacology, Faculty of Medicine, Universiti Malaya, Kuala Lumpur 50603, Malaysia

**Keywords:** *Faecalibacterium prausnitzii*, colorectal cancer, probiotics, gut microbiota, oxidative stress

## Abstract

*Faecalibacterium prausnitzii* is one of the most abundant commensals of gut microbiota that is not commonly administered as a probiotic supplement. Being one of the gut’s major butyrate-producing bacteria, its clinical significance and uses are on the rise and it has been shown to have anti-inflammatory and gut microbiota-modulating properties in the treatment of inflammatory bowel illness, Crohn’s disease, and colorectal cancer. Colorectal cancer (CRC) is a silent killer disease that has become one of the leading causes of cancer-related death worldwide. This study aimed to evaluate the anti-tumorigenic and antiproliferative role of *F. prausnitzii* as well as to study its effects on the diversity of gut microbiota in rats. Findings showed that *F. prausnitzii* probiotic significantly reduced the colonic aberrant crypt foci frequency and formation in Azoxymethane (AOM)-induced CRC in rats. In addition, the administration of *F. prausnitzii* lowered the lipid peroxidation levels in the colon tissues. For in vitro 3-[4,5-dimethylthiazol-2-yl]-2,5-diphenyl tetrazolium bromide (MTT) assay, the cell-free supernatant of *F. prausnitzii* suppressed the growth of HCT116 colorectal cancer cells in a time/dose-dependent manner. 16S rRNA gene sequencing using rat stool samples showed that the administration of *F. prausnitzii* modulated the gut microbiota of the rats and enhanced its diversity. Hence, these findings suggest that *F. prausnitzii* as a probiotic supplement can be used in CRC prevention and management; however, more studies are warranted to understand its cellular and molecular mechanisms of action.

## 1. Introduction

Colorectal cancer (CRC) is a leading cause of cancer-associated deaths globally [1]. It is believed that various factors, such as lifestyle and genetic mutations, can contribute to its development [2]. Although there are various treatments available for CRC, the patient’s survival rate is usually low [3]. The main risk factors for CRC are lifestyle choices that can lead to weight gain and diabetes. High alcohol consumption, excessive red meat consumption, and smoking are also known to contribute to the development of CRC in older individuals, but now younger individuals (less than 40 years old) are increasingly diagnosed with CRC [4,5].

During ancient times, people who suffered from gastrointestinal tract disorders were given food that contained probiotics. This food was then used by traditional healers to treat their ailments [6]. Probiotics are commonly used as a supplement for treating various diseases and conditions such as inflammatory bowel illness, Crohn’s disease, and CRC [7]. Probiotics are “live microorganisms that provide health benefits to the host when provided in the proper and adequate amounts” [8]. Currently, there are studies that support the use of probiotics for colon cancer prevention and treatment [9,10,11,12]. Even though there are many advanced medical procedures available for treating CRC, the prognosis is usually poor, and the treatment is associated with severe side effects. Probiotics have been shown to reduce some of these side effects of various treatments [13].

The composition of the human gut microbiota can affect the development of CRC. It has been shown that the presence of certain bacteria in the gut can increase or decrease the risk of colon cancer [14]. Despite the link between dysbiosis and CRC pathogenesis, several studies have shown a potential link between the gut bacterial community and the disease [15,16,17]. The presence of beneficial bacteria in the gut can confer health benefits to the host. In addition, some studies indicated the regulation of the gut microbiota can prevent the development of CRC [18].

*Faecalibacterium prausnitzii* is amongst the abundant bacterial species that can be found in the human gut [19]. It has been known to promote gut health. Although some studies have characterized the strains of this species, it is not yet clear which factors contribute to the bacterial abundance in the gut [20]. Further, its distribution is different among patients with different gut disorders. This bacteria’s abundance is seen to be reduced in various gut disorders such as Crohn’s disease, ulcerative colitis, and inflammatory bowel disease. It has been hypothesized that this finding could serve as a biomarker for gut diseases [21].

*F. prausnitzii* is a major producer of butyrate in the gut. It has anti-inflammatory properties and is known to reduce the inflammation in the intestine [22]. Butyrate is a substance found in the intestine and it has properties that protect the colon from cancer and inflammatory bowel diseases. Butyrate can also reduce the inflammation in the gut and inhibit the growth of colon cancer cells [23]. The amount of *F. prausnitzii* in the gut microbiota is influenced by factors such as pH and the oxygen content in the colon [24]. Furthermore, one study reported that *F. prausnitzii* counts reduced as a result of poor diet and smoking as well as the use of specific medicines [19].

*F. prausnitzii* was also observed to inhibit the activation of NF-kB by IL-10 in Caco-2 cells. The anti-inflammatory properties of *F. prausnitzii* can trigger pro-inflammatory cytokines and increase the production of anti-inflammatory IL-10 [25]. In addition, *F. prausnitzii* can improve the gut barrier’s functionality and reduce inflammation by delivering metabolites that are known to stimulate an immunological response [26]. Hence, in this study, we investigated the preventive role of *F. prausnitzii* probiotics against CRC initiation in azoxymethane-induced CRC in a rat model.

## 2. Materials and Methods

### 2.1. Bacterial Strains and Growth Conditions

*Faecalibacterium prausnitzii* (DSM 17677) was purchased from the Leibniz Institute DSMZ-German collection of Microorganisms and Cell Cultures GmbH, Braunschweig, Germany. The bacteria from the source came in freeze-dried form and were placed in a double-glass vial for storage. The bacteria were cultivated in reinforced clostridial medium (RCM) broth (CM0149 Oxoid, Basingstoke, Hants, UK) for three days at 37 °C in an anaerobic chamber. After that, the probiotic log-phase cultures were centrifuged at 5000× *g* and 4 °C for 15 min, and the *F. prausnitzii* cell pellets were suspended in PBS at a concentration of 1× 10^9^ colony-forming units (CFU). The cell-free supernatants were filtered through a 0.22 µm polyethersulfone (PES) membrane and kept at −20 °C in a single use aliquot until further use. 

### 2.2. HCT116 Cell Line Culture

The human colon cancer cell line HCT116 was obtained from ATCC, VA, USA. HCT116 cells were grown in Dulbecco’s modified Eagle’s medium (DMEM) at 37 °C in a humidified environment with 5% CO_2_. The media were supplemented with antibiotics and 10% fetal bovine serum (FBS).

### 2.3. MTT Assay

The MTT cell viability assay was used to determine the cytotoxicity and inhibitory effects of *F. prausnitzii* cell-free supernatant against HCT116 human CRC cells. The cells were seeded at 5000 cells per well overnight before being subjected to various concentrations of supernatant (50, 25, 12.5, 6.5, 3.13, 1.56, 0.78%) and 5-fluorouracil (5-FU) as a standard control (50, 25, 12.5, 6.5, 3.13, 1.56, 0.78 µg/mL). Then, the cells were then incubated for 12, 24, 48, and 72 h at 37 °C with 5% CO_2_ before being re-incubated with 10 µL MTT solution for 4 h. To expose the formazan crystal, the cells were lysed in 100 µL of DMSO, and the absorbance was measured with a plate reader (Perkin Elmer, Waltham, MA, USA) at 570 nm [27]. The vehicle, i.e., plain broth (RCM) and 0.1% DMSO diluted in cell culture medium (DMEM), was used as the control.

### 2.4. Animal Experiment

In total, 24 Male Sprague Dawley rats (10 weeks old) were housed in IVC cages (two to three rats per cage) in the animal experimental unit, University of Malaya, at a temperature of 22 °C and humidity of 75%, with a 12-h light/dark cycle. The rats were provided free access to Altromin rat pellets and RO water.

The rats were randomly divided into four groups of five rats each, one of which served as a normal control group. The vehicle was assigned to group 1, the control group. The animals in groups 2 to 4 were administered azoxymethane (AOM) subcutaneously at a dose of 7 mg/kg body weight once a week for three weeks to induce carcinogenesis in rats. Rats in group 3 were given oral gavage of *F. prausnitzii* (1 × 10^9^ CFU/mL) one week before AOM administration and continued for five weeks, while rats in group 4 were given 35 mg/kg of 5-fluorouracil (IP, 3 times/week) for five weeks.

### 2.5. Assessing Colon Tissue for Aberrant crypt foci (ACF)

At the end of the experiment, all rats were sacrificed under anesthesia (80 mg/kg ketamine and 5 mg/kg xylazine). Colon tissue examination was performed according to Bird’s method with slight modification [28]. The colon tissues were cut into 2-cm-square pieces and the distal and proximal pieces were placed on microscope slides with the mucosal part facing up. The tissues were fixed in formalin (10% phosphate buffered) for 24 h at room temperature, and then the colon samples were stained with 0.5% methylene blue solution for 3 min [29]. Then, the slides were rinsed using RO water until all excess dye was washed off. The total number of ACF per 2 cm segments was counted using a microscope with an attached camera. The aberrant crypts were recognized from the surrounding normal crypts by their enlarged size and visible peri cryptal zone, as well as their extended distance from the lamina to basal surface of cells. The ACF formation and multiplicity were determined for each colon segment [30].

### 2.6. Colon Histological Assessment

The colon tissues were harvested and cut into 1 × 1 cm squares, which were subsequently fixed in 10% buffered formalin. The tissues were processed using a computerized tissue preparing/processing apparatus and embedded in paraffin wax as per standard methods. The colon slices were cut into 5 μm thickness lateral to the muscularis mucosa and stained with hematoxylin and eosin stain All ACF identified in the colon tissues were examined for their features and compared to those of the normal control group under a light microscope.

### 2.7. Measurement of Oxidative Stress and Lipid Peroxidation

To remove as much blood as possible, colon specimens were immediately rinsed with cold phosphate-buffered saline (PBS). Using a tissue homogenizer, colon tissues (10% *w*/*v*) were homogenized in cold PBS (pH 7.4). Centrifugation at 4500 rpm for 15 min at 4 °C was used to remove cell debris. After that, the Cayman protein determination kit was used to measure the protein content of the supernatant samples. Malondialdehyde (MDA) levels were determined in colon tissue homogenates from treated and untreated rats to determine the degree of lipid peroxidation using the thiobarbituric acid reactive substance (TBARS) assay (Cayman Chemical Company, Ann Arbor, MI, USA) as directed by the manufacturer. The results were expressed in µM of MDA per mg of protein.

### 2.8. DNA Extraction and 16S rRNA Sequencing

To study the effect of *F. prausnitzii* on the diversity of gut microbiota in rats, the total microbial DNA was extracted from stool samples using the E.Z.N.A.^®^ Soil/Stool DNA Kit (Omega Bio-tek, Norcross, GA, USA). 16S rRNA analysis was utilized to characterize the intestinal microbiota of rats. The primers 515F: 5′-GTGCCAGCMGCCGCGGTAA-3′ and 806R: 5′-GGACTACHVGGGTWTCTAAT-3′ were used to amplify the V4 distinct region of bacterial 16S rRNA as a template. PCR was used to amplify the marker region of the bacteria according to the following protocol: 95 °C for 2 min, followed by 25 cycles at 95 °C for 30 s, 55 °C for 30 s, and 72 °C for 30 s and a final extension at 72 °C for 5 min [31,32].

The raw FASTQ 16S rRNA gene sequencing reads were demultiplexed and quality-filtered using QIIME software (version 1.9.1; http://qiime.org/scripts/assign_taxonomy.html, accessed on 15 April 2021) based on overlapping relationships; paired-reads were merged into a single read. The filtering was carried out according to the following criteria, the 300-bp reads were truncated at any site with an average quality score < 20 over a 50-bp sliding window, and truncated reads that were shorter than 50 bp were discarded. Exact barcode matching, two nucleotides mismatch in primer matching, and reads containing ambiguous characters were removed, and only sequences that overlapped longer than 10 bp were assembled according to their overlapping sequence. Reads that could not be assembled were omitted. The merged reads were used for OTU clustering, taxonomy classifying, and community diversity assessing, which were clustered with 97% similarity cut-off using UPARSE (version 7.1; http://drive5.com/uparse/, accessed on 15 April 2021), and chimeric sequences were identified and removed using UCHIME. The microbial community was used to compare the similarity or dissimilarity between different groups and the relationship between the microbial community and environmental factors, phylogenetic analysis, and alpha diversity analysis. The taxonomy of the 16S rRNA gene sequence was analyzed by RDP Classifier (Release 11.1; http://rdp.cme.msu.edu/, accessed on 15 April 2021) against the Silva (SSU123; Release 123; http://www.arb-silva.de, accessed on 15 April 2021) 16S rRNA database using a confidence threshold of 0.7. In addition, other databases were also used to compare the similarity including Greengenes (Release 13.5; http://greengenes.secondgenome.com/, accessed on 15 April 2021) and the Functional Gene Database (FGR; Release 7.3; http://fungene.cme.msu.edu/, accessed on 15 April 2021).

### 2.9. Statistical Analysis

Statistical analysis was performed using SPSS (Statistical Package for the Social Sciences) version 27.0 for Windows. The statistical significance of the data was analyzed using one-way analysis of variance (ANOVA) with Tukey’s multiple comparisons post hoc test. All the data were expressed as the mean ± standard error mean (SEM) of three replicates for the in vitro parameters and five rats for the in vivo experiment; *p* values < 0.05 were considered significant.

## 3. Results

### 3.1. F. prausnitzii Cell-Free Supernatant Inhibits Cancer Cell Proliferation

MTT assay was used to evaluate the quantity of viable HCT116 cells after exposure to *F. prausnitzii* cell-free supernatant at different time points (Figure 1A). The supernatant of *F. prausnitzii* decreased the viability of HCT116 cells in a dose-dependent manner, and the highest inhibition of cell proliferation was reported after 72 h. In comparison to *F. prausnitzii*, 5-FU was more potent in inhibiting HCT116 cell proliferation (Figure 1B). In addition, after 72 h of treatment, 5-FU exhibited a lower IC_50_ (0.84 ± 0.020 µg/mL) against HCT116 cells, while the IC_50_ of *F. prausnitzii* was 8.3 ± 0.954 % after 72 h.

### 3.2. F. prausnitzii Reduces ACF Formation in Rats

The body weight of the rats steadily increased throughout the course of the treatment except for rats in the 5-FU group, which showed a sharp decrease in week four, which could be one of the side effects of the 5-FU treatment. Although there was some difference in body weight between the groups, the differences were not statistically significant (Figure 2A).

The ACF in the colon was counted using a microscope, and the total number of crypts was recorded. The distal part of the colon showed more aberrant crypt foci (ACF) than the proximal colon. The rats supplemented with *F. prausnitzii* showed a significantly (*p* < 0.001) lower total ACF count (14.8 ± 3.86) than the AOM-control group (33.9 ± 3.18) (Table 1 and Figure 2B). The percentage ACF inhibition in the *F. prausnitzii* group was 56.3%, compared to 67.5% in the 5-FU group (Figure 2C).

### 3.3. Measurement of Oxidative Stress and Lipid Peroxidation

The malondialdehyde (MDA) levels increased in the colon tissue of AOM-induced rats, and the average concentration was 5.697 ± 1.947 µM/mg protein compared to the untreated normal control group (1.22 ± 0.473 µM/mg protein), whereas supplementation of probiotic and 5-FU treatment reduced the oxidative stress in *F. prausnitzii* and 5-FU-treated rats, and the average values of MDA were 3.868 ± 1.031 and 1.370 ± 0.371 µM/mg protein, respectively. There were no significant differences in MDA levels between the groups (Figure 2D).

### 3.4. Microscopical and Histological Findings

Methylene blue staining revealed that the rats injected with AOM developed identifiable ACF in the colon with multiple foci. There were no aberrant crypts in the colon of the rats in the normal control group. On the other hand, rats supplemented with *F. prausnitzii* showed reduced ACF numbers and less multiplicity than those in the AOM carcinogenic group (Figure 3A).

Histological examination of colon tissues revealed dysplastic and hyperplastic crypts in the AOM control group; these ACF had a larger and longer mucosal lining, visible cell corrosion, increased inflammation, nucleus crowding, loss of goblet cells, and loss of polarity. The existence of ACF-propagating mucosal glands in AOM-induced rats was distinguished by architectural atypia, elongated stratified nuclei, atypical epithelial cells, frequent mitoses and mucin degradation, and larger than normal crypts in the colonic tissues compared to normal rat colonic tissues. As compared to the AOM carcinogenic control group, all of these abnormal features were markedly reduced in the colon sections of treated rats with *F. prausnitzii* and 5-FU (Figure 3B).

### 3.5. F. prausnitzii Influences Microbial Diversity of the Gut Microbiota in Rats

The diversity of the intestinal microbiota in rats was analyzed using 16S rRNA sequencing. In total, 442 species were identified in the fecal samples of the rats, and these species belong to 358 genera and 33 phyla. Figure 4 shows the most abundant phyla (average abundances ≥ 2%) among the treated and untreated groups, which mostly accounted for at least 97% of the reads in each sample, including Firmicutes, Bacteroidota, Actinobacteriota, Proteobacteria, Verrucomicrobiota, and Patescibacteria. Across all groups, bacteria from the Firmicutes phyla were the most abundant; however, there was no significant difference in the quantity of the other phyla across all groups.

At the genus level, the top 10 genera were Romboutsia, Lactobacillus, Bifidobacterium, Bacteroides, Escherichia, Coriobacteriaceae, Staphylococcus, Ligilactobacillus, Muribaculaceae, and Akkermansia. The normal control group had the most diversity followed by the *F. prausnitzii*-treated group, 5-FU-treated group, and AOM-control group (Figure 5 and Figure 6). The Shannon diversity index is used to measure the diversity and evenness of species in a microbial community. A lower Shannon index value indicates higher diversity while a higher value means less diversity. The Shannon curve tended to plateau, indicating an increase in the sequencing volume; however, there was no significant differences in the Shannon diversity index between groups (Figure 7). There was a significant increase in the abundance of Bacteroidetes (Figure 8A) and a significant decrease in the abundance of Firmicutes (Figure 8A) in the rats treated with *F. prausnitzii* compared to the AOM control group, which was reflected in a low Firmicutes/Bacteroidetes ratio (Figure 8C).

## 4. Discussion

The majority of CRC patients are diagnosed in later stages, especially metastatic stages, lowering the patient overall survival to 10% [33]. On the other hand, all of the main CRC treatments, including surgery, chemotherapy, and radiotherapy, significantly diminish the patient’s quality of life. In light of these considerations, more effective preventative measures are needed to overcome CRC [34].

Probiotic use as a means of maintaining health and alleviating disease symptoms, particularly gastrointestinal problems, has exploded in popularity around the world [35]. Mostly probiotics of the Lactobacillus or Bifidobacterium genus and formulations containing their species are used as supplements or clinically as adjuvant therapy for different gastrointestinal diseases.

CRC is the third most prevalent cancer in men and the second most frequent cancer in women in the globe [36]. Although people with early-stage of CRC have a 5-year survival rate of 63%, CRC is the second highest cause of cancer mortality [36].

In this study, we focused on *F. prausnitzii* probiotics and assessed their ability to prevent the aberrant crypt foci precancerous lesions from developing into colorectal cancer.

*F. prausnitzii* is a metabolically versatile microbe, which may account for its abundance and wide distribution in the human gut microbiota. So far, only two phylo-groups have been recognized within this species, but the real diversity of the genus is still unknown [23]. *F. prausnitzii* is an important commensal bacterium that may exert a health benefit effect, but this species is immensely sensitive to fluctuations in the gut environment, which could limit its abundance, particularly in diseased guts [21]. *F. prausnitzii* has been identified as a major butyrate producer in the gut. Butyrate helps decrease inflammation in the intestinal mucosa and thus plays a major role in the preservation of the human gut. Previous studies on *F. prausnitzii* have mainly assessed the potential role of this probiotic against inflammatory bowel disease, Crohn’s disease, and ulcerative colitis [37]. There are very few studies evaluating the activity of *F. prausnitzii* against CRC formation and progression.

In our study, we evaluated the cytotoxic effects of *F. prausnitzii* supernatant against HCTT116 CRC cells using the MTT assay, and findings showed that the cell-free supernatant of *F. prausnitzii* was able to inhibit the proliferation of the HCT116 cells in a time-dependent manner. This result is similar to the one obtained in another study where *F. prausnitzii* supernatant was tested against the Caco-2 reporter cell line; it was found that the *F. prausnitzii* cell-free supernatant strongly inhibited the proliferation of Caco-2 cells by inhibiting NF-κB activation [38]. This was mainly as a result of the action of the metabolites such as butyrate, acetate, and propionate, which are the main metabolites of *F. prausnitzii* and were present in the supernatant. These secreted metabolites have been seen to modulate the NF-κB signaling pathway in cell culture experiments [39]. Another recent study showed that the supernatant of *F. prausnitzii* suppressed the proliferation and significantly promoted the apoptosis of MCF-7 breast cancer cells [40]. In addition, *F. prausnitzii* is a major producer of butyrate in the gut [22] and it is well known that butyrate has properties that protect the colon from cancer [23] and increases apoptosis in carcinogen-treated rats [41].

Using *F. prausnitzii* live cells for animal experimentation is not an easy task as the bacterium is extremely oxygen sensitive. We had to take a lot of care to ensure that the live bacteria administered to the rats were viable. AOM (7 mg/kg) delivered to rats for three weeks was able to induce ACF formation, which is one of the earliest changes that indicate neoplastic lesion development. Under the microscope after methylene blue staining, it was seen that *F. prausnitzii* significantly reduced the formation, multiplicity, and frequency of ACF after exposure to AOM compared to the rats in the AOM carcinogenic group. This inhibition activity could be due to the butyrate production by *F. prausnitzii,* which is vital for maintaining intestinal/gut integrity, equilibrium, and health, as well as being a major source of energy for colon cells [38].

Furthermore, histological analysis revealed that the ACF most likely formed in the AOM-induced group as dysplastic crypts, which are precursor lesions of CRC. They also exhibited hyperplastic crypts, which are usually the first stage of formation before dysplasia with the most severe dysplasia being in the AOM group compared to the other treated groups. 5-Florouoacil was more effective at inhibiting ACF formation and dysplasia; however, this came at a cost, as we noticed that the rats in the 5-FU group had more side effects such as hair loss, and some rats in the AOM group lost weight drastically.

In addition, *F. prausnitzii* was able to reduce lipid peroxidation in rats exposed to AOM, which is known to cause oxidative stress and can be measured by MDA levels. The results of this study revealed that the high levels of lipid peroxidation in the AOM-control group were ameliorated by the use of *F. prausnitzii*. These results indicate that exposure to AOM causes acute colonic cell damage. To our knowledge, this study is the first to use *F. prausnitzii* in an animal model to investigate its preventive activity against development of aberrant crypt foci, which are the pre-cancerous lesions that are usually precursors of CRC.

The rising popularity of probiotics has been widely attributed to their beneficial effects on the digestive system. They are known to improve quality of life and reduce the symptoms of various diseases. The consumption of probiotics can provide benefits to the gut microbiota and colon health. This alteration of the gut microbiota composition has been proven to prevent or treat colon cancer [17]. In this study, *F. prausnitzii* was seen to improve the diversity of the gut microbiota of the rats and increase the population of other beneficial microbes such as *Lactobacillus Bifidobacterium*, and *Ligalactobacillus*.

Administration of *F. prausnitzii* exerts pressure on the gut microbial population and influences the diversity and richness of gut microbiota. Rats treated with *F. prausnitzii* showed an increase in the abundance of Bacteroidetes compared to the control group, which was reflected as a low Firmicutes/Bacteroidetes ratio. Previous studies revealed that a high Firmicutes/Bacteroidetes ratio is associated with CRC [42] and is considered a relevant marker of gut dysbiosis [43]. These results show that *F. prausnitzii* administration significantly altered the microbiota composition of both the luminal and mucosal associated microbiota. Because *F. prausnitzii* is susceptible to changes in the intestinal environment, whether fecal or mucosal, *F. prausnitzii* has the potential to be used as a biomarker in the diagnosis of gastrointestinal disorders [44].

Fecal *F. prausnitzii* transplantation is now extensively explored as a therapeutic approach for dysbiosis of the gut microbiota, which has been linked to autoimmune disease, inflammation, and infectious diseases [45]. Butyrate-producing bacteria such as *F. prausnitzii* have been reported to block the transfer of bacterial endotoxins, which are linked to insulin resistance. In an *in vitro* experiment, researchers demonstrated that *F. prausnitzii* in human immune cells can reduce gut inflammation [46].

## 5. Conclusions

This study revealed that *F. prausnitzii* can be used as a therapy to modulate the diversity of gut microbiota and as a preventive measure against CRC formation and progression. The anti-tumorigenic potential of *F. prausnitzii* was evident by reducing the incidence of ACF lesions, slowing the progression of ACF, influencing the diversity of gut microbiota, and decreasing lipid peroxidation in the colon. *F. prausnitzii* could be considered a next-generation probiotic, which would be very useful in the management of colorectal cancer in the future. Further studies should be conducted to investigate the mechanism of actions of the anti-tumorigenic activity of *F. prausnitzii* and to elucidate its role in the immune response during initiation and development of CRC. In addition, RCT studies are required to assess the usage of *F. prausnitzii* as an adjuvant alongside conventional chemotherapy drugs.

## Figures and Tables

**Figure 1 biomedicines-10-01128-f001:**
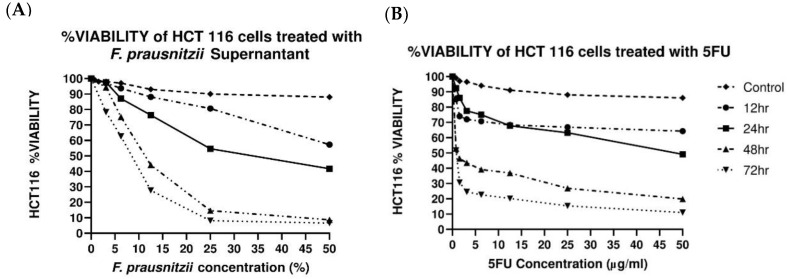
Cell viability of HCT116 colorectal cancer cells. Graphs shows time- and dose-dependent reduction of cell viability of HCT116 cells treated with *F. prausnitzii* cell-free supernatants (**A**) and 5-Fluorouracil (**B**) for 12, 24, 48, and 72 h. All values are expressed as the mean ± SEM of triplicates.

**Figure 2 biomedicines-10-01128-f002:**
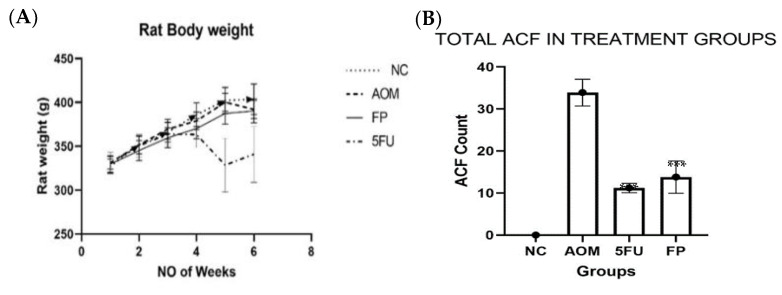
Outcomes of the animal experiment. (**A**) Body weight (gm) of rats in different experimental groups, (**B**) Total Aberrant crypt foci (ACF) count in the colon of treated and control groups, (**C**) Percentage inhibition of aberrant crypt foci (ACF) formation in colon of treated and control groups (**D**) Malondialdehyde (MDA) level in treated and untreated control groups. All values are expressed as the mean ± SEM and were analyzed using one-way ANOVA, *** *p* < 0.001 indicates a significant difference compared to the AOM group, n = 5 animals per group. (NC): Normal control, (AOM): Azoxymethane, (FP): *F. prausnitzii*, (5-FU): 5-fluorouracil.

**Figure 3 biomedicines-10-01128-f003:**
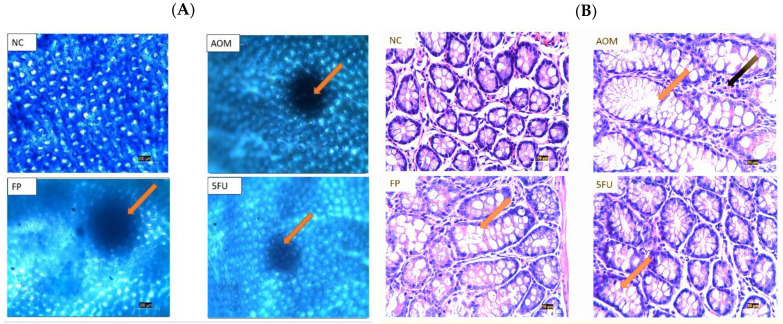
(**A**) Methylene blue stain shows the aberrant crypt foci (ACF), (scale bar= 200 µm). The arrow points to the abnormal crypts that show ACF lesions (**B**) Hematoxylin and eosin (H & E) staining findings of the colon of treated and untreated control groups, (scale bar= 50 µm). The orange arrows show enlarged crypts, while the black arrows point to the inflammatory infiltration. (NC): Normal control, (AOM): Azoxymethane, (FP): *F. prausnitzii* and (5-FU): 5-fluorouracil.

**Figure 4 biomedicines-10-01128-f004:**
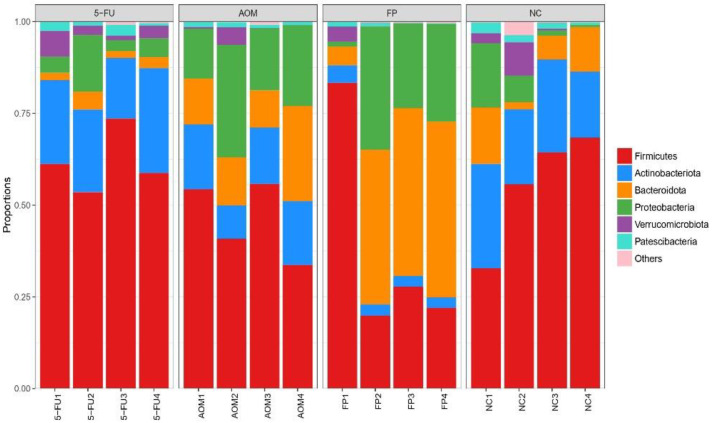
Plot of microbial diversity analysis at the phylum level. (NC): Normal control group, (AOM): azoxymethane group, (FP): *F. prausnitzii* group, (5FU): 5-fluorouracil group.

**Figure 5 biomedicines-10-01128-f005:**
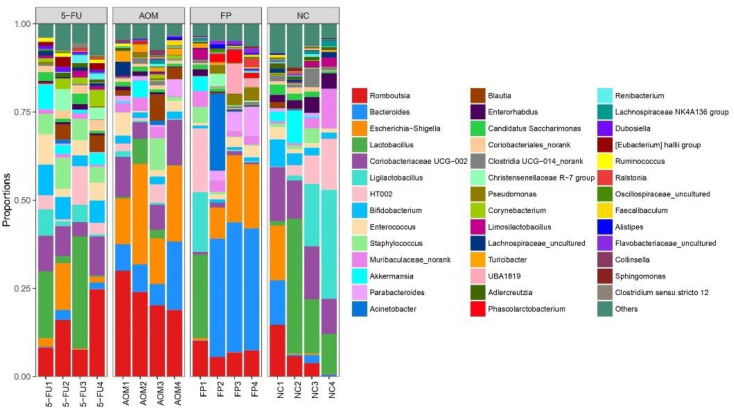
Plot of microbial diversity analysis at the genus level. (NC): Normal control group, (AOM): azoxymethane control group, (FP): *F. prausnitzii* group, (5FU): 5-fluorouracil group.

**Figure 6 biomedicines-10-01128-f006:**
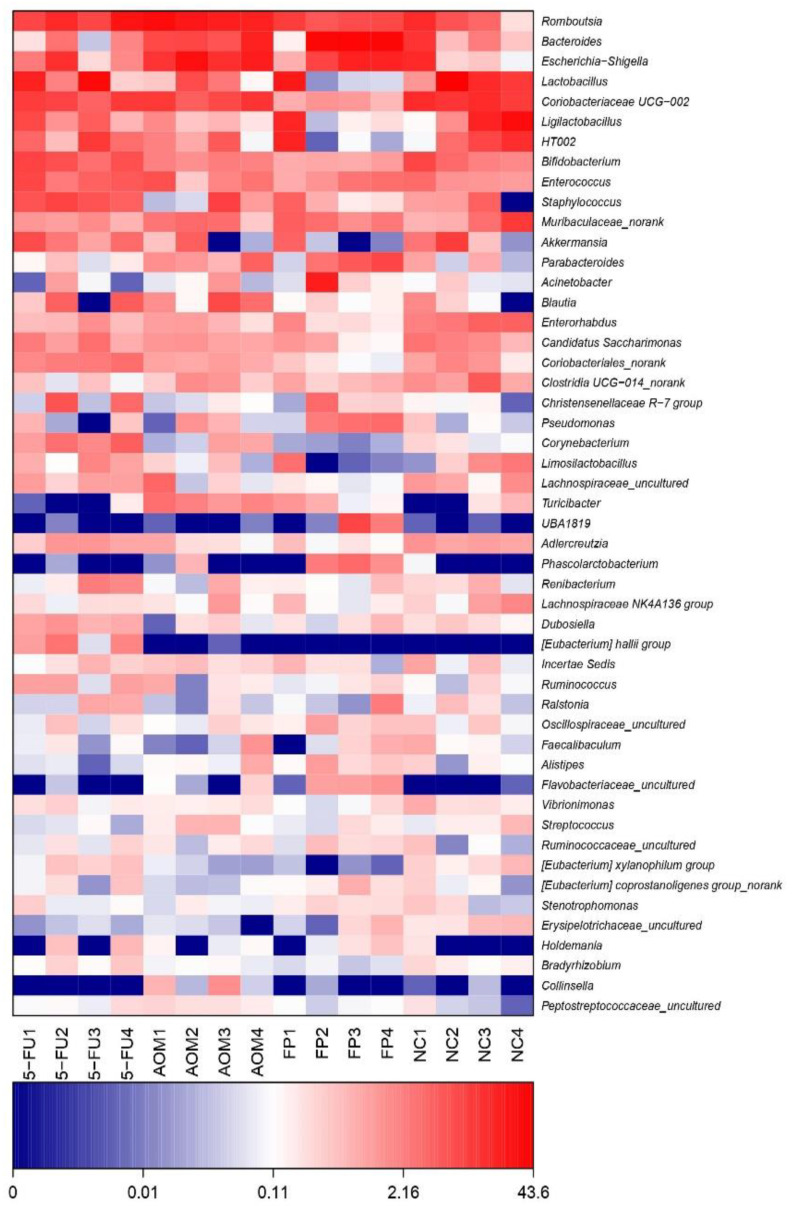
Heatmap of microbial diversity analysis at the genus level. (NC): Normal control group, (AOM): azoxymethane control group, (FP): *F. prausnitzii* group, (5FU): 5-fluorouracil reference group.

**Figure 7 biomedicines-10-01128-f007:**
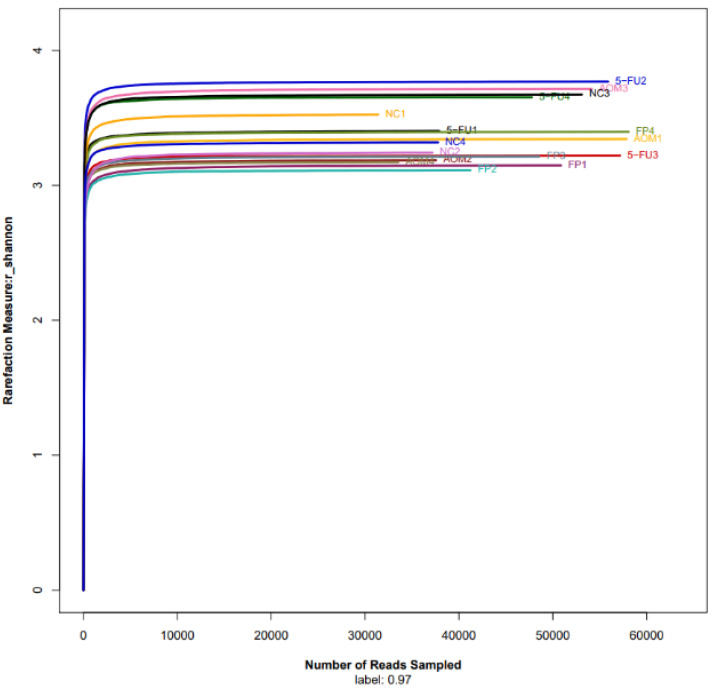
Rarefaction curve of the Shannon index shown as sequences per sample. The x-axis shows the number of reads per sample, while the y-axis indicates the Shannon index. Each curve represents a different sample and is illustrated in a different color. A lower Shannon index indicates higher diversity while higher values indicate less diversity. The microbiota of the control (NC) samples was significantly less diverse than that in the other groups. (NC): Normal control group, (AOM): azoxymethane control group, (FP): *F. prausnitzii* group, (5FU): 5-fluorouracil reference group.

**Figure 8 biomedicines-10-01128-f008:**
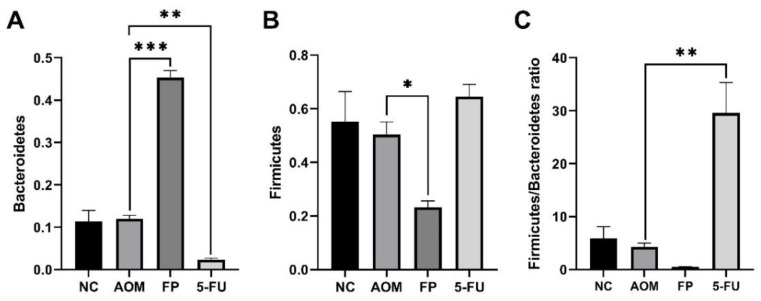
The relative abundance (%) of Bacteroidetes (**A**), (%) Firmicutes (**B**), and Firmicutes/Bacteroidetes ratio (**C**) in the treated groups and control. All values are expressed as the mean ± SEM and were analyzed using one-way ANOVA; * *p* < 0.05, ** *p* < 0.01, *** *p* < 0.001 indicate significant difference compared to the AOM group. (NC): Normal control group, (AOM): azoxymethane control group, (FP): *F. prausnitzii* group, (5FU): 5-fluorouracil reference group.

**Table 1 biomedicines-10-01128-t001:** Aberrant crypt foci (ACF) count in treated and untreated groups.

Groups	Total ACF	% Inhibition
Normal control	0	0
AOM group	33.9 ± 3.18	0
*F. prausnitzii* group	14.8 ± 3.86 ***	56.3 ***
5-FU group	11.0 ± 1.10 ***	67.5 ***

Values are expressed as the mean ± SEM, values with *** (*p* < 0.001) indicate a significant difference from the AOM group.

## Data Availability

The 16S rRNA data were uploaded to NCBI, Bio Sample database, with BioSample accessions: SAMN25826982, SAMN25826983, SAMN25826984, SAMN25826985, SAMN25826986, SAMN25826987, SAMN25826988, SAMN25826989, SAMN25826994, SAMN25826995, SAMN25826996, SAMN25826997, SAMN25826998, SAMN25826999, SAMN25827000, and SAMN25827001. Web link: https://www.ncbi.nlm.nih.gov/bioproject/805078, accessed on 7 April 2022.

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
