# Peer review of "Faecalibacterium prausnitzii Ameliorates Colorectal Tumorigenesis and Suppresses Proliferation of HCT116 Colorectal Cancer Cells"

_biomedicines, 2022, doi:10.3390/biomedicines10051128_

Round 1

Reviewer 1 Report

Dear authors,

The quality of the study was improved by last time I read.

On other hand in my opinion, some points should be addressed by authors

1) the method sessions should be improved. The method description should permit other scientists to replicate your work and more description of bioinformatic methods are lacking. No sufficient details about methods of the bioinformatic analysis were written or described (methods applied for generating figure 4-5-6).

Some examples:

a) "... quality-filtered using QIIME software (version 1.9.1)". How? Quality filters were based on the mapping quality? on the sequencing quality? Which cut-off were used? ....
b) "... OUT clustering" There are a lot clustering algorithms, which type of clustering Did authors use?
c) ".... The microbial community was used to compare the similarity or dissimilarity..." In this case reference or more details are necessary to understand the method.

2) Figure 6 b: the colour of treatment of hematoxylin and eosin seems always too near to blue colour. Could the authors explain to me, why? Maybe is it a problem with image upload?

3) Authors reported: "Administration of F. prausnitzii exerts pressure on the gut microbial population and influences the diversity and richness of gut microbiota. There was an increase in the abundance of Bacteroidetes in the rats treated with F. prausnitzii compared to other group".

To support this sentence, authors should add a statistical test (also only for Bacteroidetes beetween different groups), also because the shannon index (considering bacteria population overall) is not significantly different in the groups.

Author Response

Thank you for your prompt handling of our recently submitted manuscript, with ID “biomedicines-1702690” entitled “Faecalibacterium prausnitzii ameliorates colorectal tumorigenesis and suppresses proliferation of HCT116 colorectal cancer cells”.

The response to the reviewers’ comments is as given below and the corrections are highlighted in the revised manuscript.

Reviewer 1

1) the method sessions should be improved. The method description should permit other scientists to replicate your work and more description of bioinformatic methods are lacking. No sufficient details about methods of the bioinformatic analysis were written or described (methods applied for generating figure 4-5-6).

Some examples:

a) "... quality-filtered using QIIME software (version 1.9.1)". How? Quality filters were based on the mapping quality? on the sequencing quality? Which cut-off were used? ....

Thank you, we have improved the methods of the bioinformatic analysis and more description has been added as follows:

“The raw FASTQ 16S rRNA gene sequencing reads were demultiplexed, and quali-ty-filtered using QIIME software (version 1.9.1; http://qiime.org/scripts/assign_taxonomy.html) based on overlap relationship, paired-reads were merged in to a single read. The filtering was carried out according to the following criteria, the 300 bp reads were truncated at any site receiving an average quality score <20 over a 50 bp sliding window, and discarding the truncated reads that were shorter than 50 bp. Exact barcode matching, 2 nucleotide mismatch in primer matching, reads containing ambiguous characters were removed and only sequences that overlap longer than 10 bp were assembled according to their overlap sequence. Reads which could not be assembled were left out. The merged reads were used for OTU clustering, taxonomy classifying and community diversity assessing, which were clustered with 97% similarity cut-off using UPARSE (version 7.1; http://drive5.com/uparse/) and chimeric sequences were identified and removed using UCHIME. The microbial community was used to compare the similarity or dissimilarity between different groups, the relationship between microbial community and environmental factors, phylogenetic analysis, and alpha diver-sity analysis. The taxonomy of 16S rRNA gene sequence was analyzed by RDP Classifier (Release 11.1; http://rdp.cme.msu.edu/) against the Silva (SSU123; Release 123; http://www.arb-silva.de) 16S rRNA database using confidence threshold of 0.7. In addi-tion, other databases were also used to compare the similarity including Greengenes (Re-lease 13.5; http://greengenes.secondgenome.com/) and Functional gene database (FGR; Release 7.3; http://fungene.cme.msu.edu/).”

b) "... OUT clustering" There are a lot clustering algorithms, which type of clustering Did authors use?

We have used the OUT clustering algorithm UPARSE (version 7.1; http://drive5.com/uparse/) as explained above. A new paragraph has been added under section 2.8, page 4.

c) ".... The microbial community was used to compare the similarity or dissimilarity..." In this case reference or more details are necessary to understand the method.

We have added more details on the various databases that were used to compare the similarity or dissimilarity between different groups, page 4.

2) Figure 6 b: the colour of treatment of hematoxylin and eosin seems always too near to blue colour. Could the authors explain to me, why? Maybe is it a problem with image upload?

The colour of the H & E staining images are near to blue colour because we had an issue with the setting of the software for microscope image acquisition. This has been settled in the revised manuscript (Figure 3 B).

3) Authors reported: "Administration of F. prausnitzii exerts pressure on the gut microbial population and influences the diversity and richness of gut microbiota. There was an increase in the abundance of Bacteroidetes in the rats treated with F. prausnitzii compared to other group".

To support this sentence, authors should add a statistical test (also only for Bacteroidetes beetween different groups), also because the shannon index (considering bacteria population overall) is not significantly different in the groups.

We have performed a statistical analysis for the Firmicutes/Bacteroidetes ratio between treated and control groups. A new figure has been added to show the statistical differences of Firmicutes/Bacteroidetes ratio among groups (Figure 8, page 10).

Reviewer 2 Report

The authors have reached all my queries.

Author Response

Thank you for the positive feedback.

Round 2

Reviewer 1 Report

Dear authors,

The quality of the work was improved.

Minor:

Please add the type of statistical test was performed in the figure 8.  (* p < 90.05, ** p < 0.01, *** p < 0.001 indicate significant difference compared to AOM group.)

Author Response

Thank you for the suggestion. We have used one-way ANOVA for this test and this has been added to the legend of figure 8.
